# From Values to Opinions: Predicting Human Behaviors and Stances Using Value-Injected Large Language Models

Dongjun Kang[1]  Joonsuk Park[2*]  Yohan Jo[3*]  JinYeong Bak[1*]

[1]Sungkyunkwan University, Suwon, South Korea
[2]University of Richmond, VA, USA
[3]Seoul National University, Seoul, South Korea
ehdwns2356@skku.edu, park@joonsuk.org,
yohan.jo@snu.ac.kr, jy.bak@skku.edu

## Abstract

Being able to predict people's opinions on issues and behaviors in realistic scenarios can be helpful in various domains, such as politics and marketing. However, conducting large-scale surveys like the European Social Survey to solicit people's opinions on individual issues can incur prohibitive costs. Leveraging prior research showing influence of core human values on individual decisions and actions, we propose to use value-injected large language models (LLM) to predict opinions and behaviors. To this end, we present Value Injection Method (VIM), a collection of two methods—argument generation and question answering—designed to inject targeted value distributions into LLMs via fine-tuning. We then conduct a series of experiments on four tasks to test the effectiveness of VIM and the possibility of using value-injected LLMs to predict opinions and behaviors of people. We find that LLMs value-injected with variations of VIM substantially outperform the baselines. Also, the results suggest that opinions and behaviors can be better predicted using value-injected LLMs than the baseline approaches.[1]

## 1 Introduction

The ability to reliably predict people's opinions on particular issues or how they would choose to behave in different real-life scenarios can be beneficial to numerous professionals, including politicians and marketers. To this end, there exist large-scale surveys soliciting opinions on various issues, such as European Social Survey (ESS).[2] However, collecting opinions on individual issues in this way is laborious and costly.

Luckily, studies on human values claim that people have a small set of core values, which affects the daily decisions and actions (Stern et al., 1999; Bardi and Schwartz, 2003). For instance, the Schwartz value theory (Schwartz et al., 2012) specifies ten values—such as security and achievement—that are central to human life. Since these values are more manageable to collect from people than their opinions on every issue of our interest, we seek to predict people's opinions and behaviors based on their core values. More specifically, we propose to inject a target value distribution to large language models (LLMs) and have them predict the opinions and behaviors of people with similar value distributions.

From a technical perspective, LLMs are pre-trained on large corpora and thus inherently lack personality (Huang et al., 2022). This is not only problematic for our application, but also for others like chatbots, where LLMs with particular personalities are desired. To this end, researchers have measured cultural values embedded in LLMs (Arora et al., 2023) and investigated methods that simulate human behaviors (Aher et al., 2023), among others. However, to the best of our knowledge, there has not been an attempt to inject a full set of human values into LLMs and use it for predicting opinions and behaviors of people with similar value distributions.

In this paper, we propose the **V**alue **I**njection **M**ethod (VIM) for injecting specific value distributions into LLMs. VIM consists of *argument generation* (AG) and *question answering* (QA). The AG method aims to inject values by training LLMs to generate opinions on issues consistent with the targeted value distribution. The QA method, on the other hand, trains LLMs to specify how similar they are to a given description of a person, on a 6-point scale from "Not like me at all" and "Very much like me."

We first verify the effectiveness of VIM (Section 5). We inject values into LLAMA (Touvron et al., 2023) using variations of VIM resulting in three value-injected LLMs: VILLAMA, VIL-

---

*Corresponding authors
[1]Code: https://github.com/dongjunKANG/VIM
[2]https://www.europeansocialsurvey.org/

LAMA$_{AG}$, and VILLAMA$_{QA}$. Then, we test their performances against prompt-based baselines on two tasks: value survey and argument generation. The experiment results demonstrate that LLMs trained via VIM outperform the baselines on both tasks and that the variation of VIM using both methods is superior.

We then test value-injected LLMs' ability to predict human opinions and behaviors (Section 6). In particular we investigate the following questions: *Can a value-injected LLM predict the behavior of a person with the same value distribution in a realistic situation?* and *Can a value-injected LLM predict the stance of a person with the same value distribution on political, social and other issues?* The experiment results show that the answers to both questions are true to a degree. In the behavior prediction task, predictions of VILLAMA show a substantial alignment to the gold standard behaviors, achieving an average of 0.071 normalized mean squared error (NMSE). In the opinion prediction task, VILLAMA achieves 0.099 NMSE, significantly outperforming the baselines ranging from 0.137 to 0.221.

Our contributions are threefold:

- We propose the novel problem of predicting human behaviors and opinions with specific values.

- We present Value Injection Method (VIM), an effective method for injecting desired values into LLM.

- We demonstrate that value-injected LLMs outperform the baselines in predicting the behaviors and opinions of people who have similar value distributions to their target value distributions.

## 2 Related Work

The Schwartz theory of basic values identifies ten basic human values that serve to characterize people's attributes:

- **Achievement (Ach):** Personal success through demonstrating competence according to social standards.

- **Benevolence (Ben):** Preserving and enhancing the welfare of those with whom one is in frequent personal contact.

- **Conformity (Con):** Restraint of actions, inclinations, and impulses likely to upset or harm others and violate social expectations or norms.

- **Hedonism (Hed):** Pleasure or sensuous gratification for oneself.

- **Power (Pow):** Social status and prestige, control or dominance over people and resources.

- **Security (Sec):** Safety, harmony, and stability of society and relationships.

- **Self-Direction (SD):** Independent thought and action–choosing, creating, exploring.

- **Stimulation (Sti):** Excitement, novelty, and challenge in life.

- **Tradition (Tra):** Respect, commitment, and acceptance of the customs and ideas that one's culture or religion provides.

- **Universalism (Uni):** Understanding, appreciation, tolerance, and protection for the welfare of all people and for nature.

The Schwartz value theory is an appropriate framework for representing human personality in our study. It provides a comprehensive understanding of individuals and groups by considering multiple values. For example, research has shown that Chinese shopper tourists make purchases of items aligned with specific values, such as passion and jewelry (Choi et al., 2016). Additionally, people's prioritized values play a role in their political decisions, including voting (Sagiv and Schwartz, 2000; Caprara and Zimbardo, 2004). Furthermore, Bonetto et al. (2021) explored the relationship between values and people's opinions regarding movement restrictions and social distancing measures in the context of COVID-19.

Personality theories, which seek to comprehend human behavior and cognition, have been employed in the realm of Natural Language Processing (NLP) research. Lately, there has been an escalating interest in exploring the utilization of these theories within generative language models, with the purpose of generating sentences that closely resemble human-like language. Those studies aimed to identify the MBTI type and human value scale of LLMs by prompting them to answer questionnaires like the MBTI questionnaire or Portrait Values Questionnaire (PVQ) (Rao et al., 2023; Miotto et al., 2022). In addition to measuring personality, there are studies that quantitatively measured whether prompting can induce desired personality traits (Jiang et al., 2023; Caron and Srivastava, 2022).

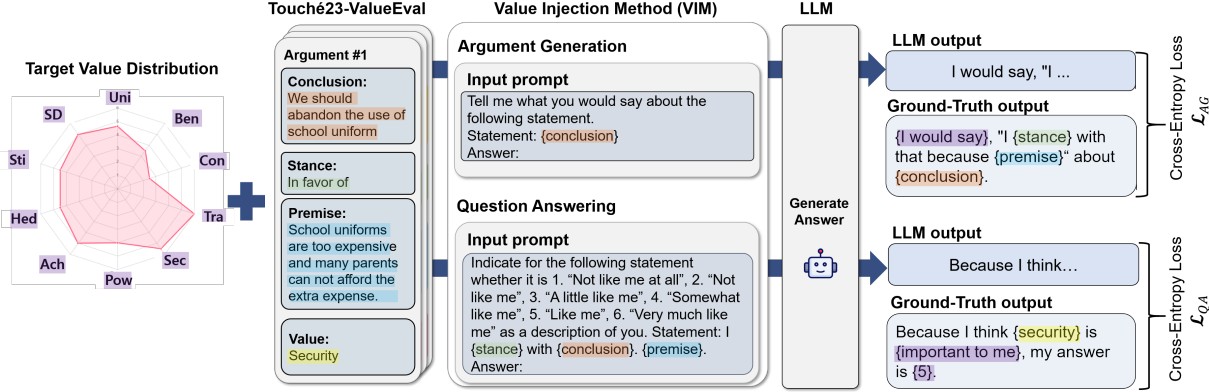

Figure 1: Overview of Value Injection Method. It consists of two methods: argument generation and question answering. Both methods utilize the Touché23-ValueEval dataset to create prompts and ground-truth answer outputs. The LLM is trained by minimizing the cross-entropy loss between the LLM's output given the input prompt and the corresponding ground-truth output.

Researchers have explored various techniques to guide language models in generating text that reflects specific personas or styles. For example, Rubin et al. (2022) conducted a study analyzing effective prompts in the in-context learning approach, which enables the generation of desired sentences without fine-tuning the model. In another study, Ouyang et al. (2022) demonstrated exceptional performance in tasks such as learning from intended instructions and mitigating the generation of toxic output by utilizing reinforcement learning with human feedback. This methodology helps align language models with human intent, thereby improving their ability to produce desired outputs. However, to the best of our knowledge, there has been no prior research investigating methods for injecting human values into LLMs.

## 3 Value Injection Method (VIM)

To inject human values into LLM, we propose the value injection method (VIM) consisting of *argument generation* (AG) and *question answering* (QA). Suppose we inject a target value distribution $V_t = \{v_t^{Ach}, v_t^{Ben}, \ldots, v_t^{Uni}\}$ into a LLM $M$, where $v_t^*$ ranges between 1 and 6 (according to PVQ). For this, we use the Touché23-ValueEval dataset (Mirzakhmedova et al., 2023) consisting of value-related arguments. Each argument $a = \{c_a, s_a, p_a, V_a\}$ consists of conclusion, stance, premise, and values: the conclusion ($c_a$) represents a specific topic, the stance ($s_a$) indicates whether it is in favor of or against the conclusion, and the premise ($p_a$) corresponds to the reasoning behind it. Each argument is labeled with values expressed in

the premise $V_a = \{v_a^{Ach}, v_a^{Ben}, \ldots, v_a^{Uni}\}$, where $v_a^*$ is 1 if the value appears in the premise and 0 otherwise. Table 12 shows an example of this dataset. We split the data with a ratio of 80:10:10 for training:validation:test.

**Argument Generation (AG)** This method injects $V_t$ into $M$ by fine-tuning $M$ to generate stances and premises that reflect $V_t$ for a given conclusion. Algorithm 1 outlines the process. At a high level, we split arguments in the dataset into two groups. The first group is arguments that are likely to be made by someone who has $V_t$, and the second group is arguments that are unlikely to be made by them. To be specific, for each argument and its values, we look at the corresponding value scores in $V_t$ and take the minimum score. If the minimum score is greater than or equal to a threshold $\gamma$, then this argument is put into the first group; otherwise, the second group. The rationale is that the likelihood of an argument being made by a person is bounded by their least prioritized value that is expressed in the argument. For the first group of arguments, the model is trained to generate "I would say [argument]", whereas for the second group, "I would not say [argument]" (see Table 16 for the exact prompts). We use the cross-entropy loss $\mathcal{L}_{AG}$ with next word prediction.

**Question Answering (QA)** In contrast to AG, QA prompts LLM $M$ to generate the stance and premise in relation to the conclusion. The possible stances for each question are based on the six options of the PVQ: "Not like me at all", "Not like me", "A little like me", "Somewhat like me", "Like me", and "Very much like me", each associated

**Algorithm 1** Argument Generation

**Input**
1: Arguments $A = \{a_1, \ldots, a_N\}$
2: Target group value distribution $V_g = \{v_g^{Ach}, v_g^{Ben}, v_g^{Con}, \ldots, v_g^{Uni}\}$
3: Large language model $M$

**Training**
1: **for** $\forall a \in A$ **do**
2:    Create an empty list $L_a$
3:    **for** $\forall v_a \in V_a$ **do**
4:     **if** $v_a^z == 1$ **then**
5:      Append $v_g^z$ into $L_a$
6:     **end if**
7:    **end for**
8:    **if** $\min(L_a) \geq \gamma$ **then**
9:     $w_a$ = 'I would say'
10:    **else**
11:     $w_a$ = 'I would not say'
12:    **end if**
13:    Make a GT argument with $c_a, s_a, p_a$, and $w_a$
14:    Make a prompt $prm_a$ with $c_a$
15:    Generate an argument from $M$ by $prm_a$
16:    Update the parameters of $M$ by $\mathcal{L}_{AG}$
17: **end for**

**Output**
1: Trained target group LLM $M_g$

---

**Algorithm 2** Question Answering

**Input**
1: Arguments $A = \{a_1, \ldots, a_N\}$
2: Target group value distribution $V_g = \{v_g^{Ach}, v_g^{Ben}, v_g^{Con}, \ldots, v_g^{Uni}\}$
3: Large language model $M$

**Training**
1: **for** $\forall a \in A$ **do**
2:    **for** $\forall v_a \in V_a$ **do**
3:     **if** $v_a^z == 1$ **then**
4:      $l_a = v_g^q \; // \; 1 + Bernoulli(v_g^z \% 1)$
5:      Make a GT answer with $z, l_a$
6:      Make a prompt $prm_a$ with $c_a, s_a, p_a$
7:      Generate an answer from $M$ by $prm_a$
8:      Update the parameters of $M$ by $\mathcal{L}_{QA}$
9:     **end if**
10:    **end for**
11: **end for**

**Output**
1: Trained target group LLM $M_g$

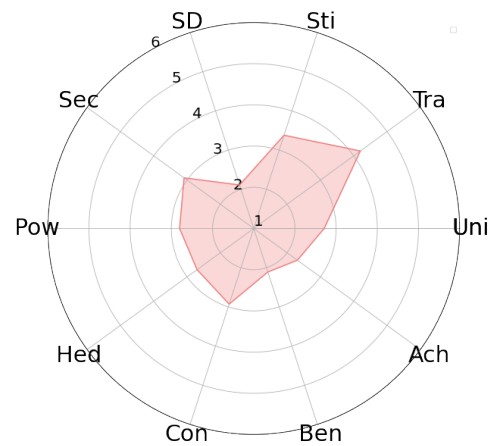

Figure 2: Example of a target group value distribution. Score of 1 indicates not consider the value at all, score of 6 correspond to the value being very important. This distribution shows relatively high tradition, simulation, and security. The value scores for this distribution are as follows: achievement=2.3, benevolence=2.1, conformity=2.9, hedonism=2.7, power=2.8, security=3.1, self-direction=2.1, stimulation=3.4, tradition=4.2, universalism=2.7.

with integers from 1 to 6, respectively.

Algorithm 2 outlines the process. We use an argument $a$ and the target value distribution $V_g$ for each iteration of training a LLM $M$. As we will see in Section 4, a target value distribution can take real numbers as value scores. To map these scores to the six options in each question $(1, 2, 3, 4, 5, 6)$, if a value score is not a whole number, we rounded it down or up to an integer probabilistically based on its fractional part (e.g., if the score is 5.2, then it is rounded to 5 with an 80% chance or to 6 with a 20% chance). We construct a ground-truth (GT) answer that first states a value $z$ associated with the argument, followed by the final choice $l_a$. This allows us to update the parameters of the LLM $M$ using the cross-entropy loss $\mathcal{L}_{QA}$ by comparing the ground-truth answer with the answer generated by $M$. Through this process, we can obtain trained $M_g$ to generate appropriate answers for value-related questions based on the target value distribution $V_g$. Please refer to Table 17 for the exact prompt format of the QA method.

## 4 Experimental Setup

### 4.1 Target Value Distributions

For thorough evaluation of VIM, we test various value distributions for injection, while ensuring that those distributions are realistic. To that end, we identified representative value distributions among humans using the European Social Survey (ESS)

| Section | Dataset | Prompt | Model Output | Ground Truth | Metric |
|---|---|---|---|---|---|
| §5.1 | Portrait Values Questionnaire | Table 18 | Number | Group value distribution | NMSE |
| §5.2 | Touché23-ValueEval | Table 19 | Premise & Stance | - | Winning ratio |
| §6.1 | VALUENET | Table 20 | Agree or Disagree | Group value distribution | NMSE |
| §6.2 | European Social Survey | Table 21 | Number | Number | NMSE |

Table 1: Summary of experiments to demonstrate that VILLAMA has the ability of reflecting the value distribution consistently in various tasks. '-' denotes the absence of ground-truth data, so we conduct a human evaluation where annotators determine the more appropriate generated argument between two different trained LLMs.

dataset. ESS is a large-scale survey conducted every two years for individuals in Europe. As part of the survey, participants answer the Portrait Values Questionnaire (PVQ) (Schwartz, 2021), a widely used questionnaire for profiling the respondent's value distribution according to Schwartz's theory. The resulting distribution is a 10-dimensional vector where each element represents the score of each value ranging between 1 (not at all) and 6 (very likely).

To identify representative value distributions from ESS, we first computed the value distributions of 54,763 people from 28 European countries based on their responses to PVQ. Next, we clustered the distributions using K-means clustering, where each data point represents each person's value distribution as a 10-dimensional vector. By applying the elbow method, we determined that 100 clusters are suitable (refer to Appendix A for more details). Lastly, we took the average of the value distributions in each cluster, resulting in 100 representative value distributions. In addition to them, we also included 28 value distributions that represent the 28 countries in ESS, by taking the average of value distributions for each country. Figure 2 shows one example of group value distribution. We train one LLM for each target value distribution and report the average score of all LLMs.

## 4.2 Models

**Value-injected LLMs (VILLAMA, VILLAMA$_{AG}$, VILLAMA$_{QA}$)** Our value-injected LLAMA (VILLAMA) is LLAMA-7B (Touvron et al., 2023) fine-tuned on value injection tasks through Low-Rank Adaptation (LoRA) (Hu et al., 2022). The total loss function is the combination of $\mathcal{L}_{AG}$ and $\mathcal{L}_{QA}$. VILLAMA$_{AG}$ and VILLAMA$_{QA}$ are variations of VILLAMA trained for an ablation study. The former is LLAMA trained with the argument generation task only, and the latter, question answering.

**Baselines (LLAMA$_{Short}$, LLAMA$_{Long}$, ChatGPT$_{Long}$)** Modern decoder-based LLMs have shown impressive in-context learning performance (Brown et al., 2020; Mishra et al., 2022). We compare the performance of VILLAMA with three zero-shot prompting baselines that receive the target value distribution in the prompt:

1. LLAMA$_{Short}$ is the pretrained LLAMA-7B. It is given the target value distribution and the task description in the prompt. Please refer to Table 22 for the exact prompt;

2. LLAMA$_{Long}$ is the same as LLAMA$_{Short}$, except the prompt now includes the definition of each value. Please refer to Table 23 for the exact prompt; and

3. ChatGPT$_{Long}$ is the same as LLAMA$_{Long}$, except ChatGPT is used in place of LLAMA.

## 4.3 Experiment Overview

We compare VILLAMA with baselines to demonstrate its ability in four tasks, as summarized in Table 1. For the evaluation of value injection itself, we test:

- how well its responses to PVQ recovers the target value distribution (Section 5.1)

- how well it generates arguments that reflect the target value distribution (Section 5.2).

For the evaluation of its ability to predict human behavior and opinions, we test:

- how well it predicts whether people with the target distribution would conduct certain behaviors or not in everyday situations (Section 6.1)

- how well its responses to questions about specific issues (e.g., political and religious topics) reflect the stance of people who have the target distribution (Section 6.2).

| Model | NMSE | Model | NMSE |
|---|---|---|---|
| LLAMA$_{Short}$ | 0.196 | VILLAMA$_{AG}$ | 0.182 |
| LLAMA$_{Long}$ | 0.189 | VILLAMA$_{QA}$ | 0.053 |
| ChatGPT$_{Long}$ | 0.146 | VILLAMA | **0.034** |

Table 2: NMSE between the model's value distribution and the target value distribution averaged across 128 target value distributions. A lower average NMSE score indicates a higher alignment between the model's value distribution and the target value distribution. The best performance is represented in **bold**, while the second best performance is represented with underlined.

# 5 Experiment 1: Value Injection

LLMs that have successfully been injected with a certain value distribution should be able to reflect the value distribution consistently in various scenarios and tasks, such as, in a value-profiling survey and argumentation.

## 5.1 Evaluation 1: Value Survey

One straightforward approach to testing the success of value injection is comparing a model's self-reported value distribution with the target value distribution injected into the model. Since PVQ (Schwartz, 2021) is the most widely used survey for measuring people's value distribution (based on the Schwartz value theory), we prompt value-injected LLMs to answer the PVQ questions. Please refer to Table 14 for example questions of PVQ.

**Setup** We created PVQ prompts for this task following the template shown in Table 18. Using these prompts, we instruct LLMs to select one of the six possible responses for each survey question these responses indicate the degree of similarity between the respondent and the description in the question, from 'Not like me at all' to 'Very much like me'. Once finished, we compute the value distribution from the responses according to the formula specified by PVQ.

We introduce a metric called Normalized Mean Squared Error (NMSE), which represents the difference between the normalized (between 0 and 1) predicted value scores $\hat{Y}_i$ and the normalized target value scores $Y_i$. Smaller NMSE indicates a closer alignment between the predicted and target value scores. The process was repeated for 128 target value distributions (obtained in §4.1) and the average is reported.

**Results** Table 2 presents the results of the PVQ evaluation. VILLAMA generates survey responses that better align with the target value distribution than other baselines. LLAMA$_{Long}$ exhibits the highest NMSE value, indicating a lack of alignment with the target value distribution. Baselines with longer prompts achieve lower errors, demonstrating the efficacy of adding value definitions in the prompt. However, the performance is still not significantly better, even for ChatGPT, which is a much larger model. Example results of this task are provided in Table 24.

With regard to the ablation study for VILLAMA, using both the AG and QA methods achieves the best performance, and simply training only with the AG method results in the worst performance. When trained using the QA method—which is similar to the PVQ task in format—the performance is better than the AG method only, but still falls behind VILLAMA. In addition, to verify the effectiveness of VIM, we adopt a paired t-test that shows how much the method affects the results. We compared the results of LLAMA$_{Long}$ and VILLAMA on value survey. For the value survey, VILLAMA's improvement over LLAMA$_{Long}$ is statistically significant ($p < 0.001$).

## 5.2 Evaluation 2: Argument Generation

For the second evaluation, we test LLMs' ability to generate arguments that reflect the target value distribution, since examining opinion is one of the ways to reveal human values (Bergman, 1998; Hoffman and Slater, 2007), and argument is a means to express opinions. Determining whether a given argument reflects a target value distribution is difficult to automate. Thus, we ask human judges to make the call.

**Setup** First, we randomly selected 40 from a total of 128 target distributions. Within each target, we sampled two conclusions from the test set of the Touché23-ValueEval dataset (see the first paragraph of Section 3 for the description of this dataset). For each conclusion, we prompted each LLM to generate a stance and a premise based on the target value distribution.

Then, three human annotators were presented with two arguments—conclusion and premise—generated by two different LLMs with prompting and VIM for the same target value distribution, and asked to determine which argument better reflects the target value distribution. When unsure, they

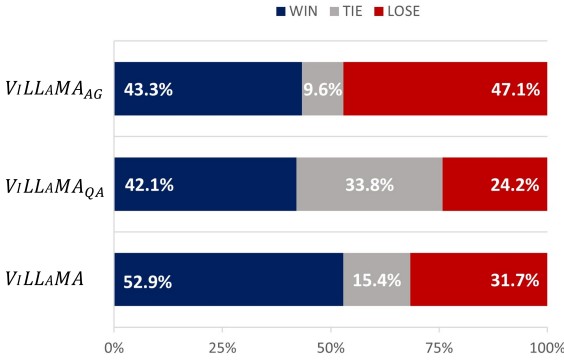

Figure 3: Human evaluation results showing the percentage of annotators selected between VɪLLaMA and LLAMA$_{Long}$. Annotators were asked to select which of the argument generated using VɪLLaMA and LLAMA$_{Long}$ is closer to the arguments generated by the target group. Among the methods, VɪL-LaMA showed the highest win ratio.

were allowed to select "I don't know." A total of 10 graduate students fluent in English served as annotators after learning the Schwartz value theory. The inter-annotator agreement, measured using Fleiss' kappa (Fleiss, 1971), among the annotators was 0.54.

**Results** Figure 3 presents the win, lose, and tie results for the variants of VɪLLaMA against LLAMA$_{Long}$. Both VɪLLaMA$_{AG}$ and VɪL-LaMA$_{QA}$ exhibit similar win ratios, but VɪL-LaMA$_{AG}$ demonstrates a higher lose ratio compared to VɪLLAMA$_{QA}$. This indicates that VɪL-LaMA$_{AG}$ generates arguments that reflect the target value distribution less effectively than VɪL-LaMA$_{QA}$ does. Note that VɪLLaMA achieves the highest win ratio, indicating that when trained for both value injection methods, the target value distribution is injected into the LLM more reliably. Example results of this task are provided in Table 12.

## 6 Experiment 2: Opinion & Behavior Predictions with Value-injected LLMs

Given value-injected LLMs, we evaluate their ability to predict human behaviors in everyday scenarios and opinions on various issues based on the underlying value distribution.

### 6.1 Behavior Prediction

In this section, we investigate the question: *Can a value-injected LLM predict the behavior of a person with the same value distribution in a realistic situation?* Schwartz (2013) examines the relationship between values and behavior in real-world situations.

**Setup** VALUENET (Qiu et al., 2022) is a dataset derived from the SOCIAL-CHEM-101 dataset (Forbes et al., 2020), which contains various behavioral patterns observed in everyday life. Each of the 21,374 scenarios in VALUENET is tagged with one value from the Schwartz value theory and specified as having a "Positive", "Negative", or "Unrelated" relationship with the given value. Please refer to Table 13 for examples of scenarios from VALUENET.

We construct a test scenario set from VALUENET by randomly selecting a total of 500 scenarios with 50 scenarios (25 positive and 25 negative) for each of the 10 values. A LLM is prompted with a test scenario and asked if it would behave the same way. It should answer either "agree" or "disagree". Please refer to Table 20 for the prompt template. The agreement ratio is the percentage of cases in which the LLM either agrees in positive scenarios or disagrees in negative scenarios, across all test scenarios. We calculated the NMSE between the re-scaled target value (ranging from 0 to 1) and the agreement ratio.

**Results** Table 3 presents the results of the behavior prediction task. Overall, VɪLLaMA generates answers that align with the target value distribution more effectively than the other baselines. This suggests that VɪLLaMA can predict human behavior in everyday life situations more accurately based on the value distribution. However, for a few values, such as Benevolence, Hedonism, and Tradition, LLAMA$_{Long}$ achieved the best performance. Paired t-test results of LLAMA$_{Long}$ and VɪLLaMA varied by value; the improvement was statistically significant for Achievement and Self-direction ($p < 0.001$), but no significance was found for the other values. Interestingly, LLAMA$_{Short}$ and LLAMA$_{Long}$ showed a lower mean error than ChatGPT$_{Long}$, even though they are smaller models and LLAMA$_{Short}$ has less information in the prompt. Example results of this task are provided in Table 26.

### 6.2 Opinion Prediction

In this section, we tackle the question: *Can a value-injected LLM predict the stance of a person with the same value distribution on political, social and other issues?* In contrast to the behavior prediction

| Model | Ach | Ben | Con | Hed | Pow | Sec | SD | Sti | Tra | Uni | Ave. |
|---|---|---|---|---|---|---|---|---|---|---|---|
| LLAMA$_{Short}$ | 0.092 | 0.180 | 0.082 | 0.091 | 0.031 | 0.138 | 0.095 | 0.065 | 0.086 | 0.179 | 0.104 |
| LLAMA$_{Long}$ | 0.075 | **0.118** | 0.057 | **0.049** | 0.030 | 0.098 | 0.091 | 0.050 | **0.069** | 0.095 | 0.073 |
| ChatGPT$_{Long}$ | 0.092 | 0.251 | 0.177 | 0.061 | 0.032 | 0.187 | **0.058** | 0.075 | 0.179 | 0.156 | 0.127 |
| VILLAMA$_{AG}$ | 0.161 | 0.247 | 0.106 | 0.123 | 0.044 | 0.206 | 0.082 | 0.097 | 0.140 | 0.151 | 0.137 |
| VILLAMA$_{QA}$ | 0.065 | 0.127 | **0.054** | 0.053 | **0.029** | 0.101 | 0.075 | 0.058 | 0.084 | 0.109 | 0.075 |
| VILLAMA | **0.049** | 0.125 | 0.056 | 0.052 | 0.035 | **0.097** | 0.074 | **0.049** | 0.072 | **0.094** | **0.071** |

Table 3: Results for the behavior prediction task. The difference between the model and the target group's real-life behaviors was quantified using normalized mean squared error (NMSE), where lower values indicate a better prediction of the target group's behavior. The best performance is represented in **bold**, while the second best performance is represented with underlined. VILLAMA (ours) outperforms other baselines.

| Model | MST | PSWB | POL | UD | Ave. |
|---|---|---|---|---|---|
| LLAMA$_{Short}$ | 0.197 | 0.148 | 0.384 | 0.153 | 0.221 |
| LLAMA$_{Long}$ | 0.079 | 0.115 | 0.281 | 0.072 | 0.137 |
| ChatGPT$_{Long}$ | 0.222 | 0.133 | **0.106** | 0.216 | 0.169 |
| VILLAMA$_{AG}$ | 0.106 | 0.078 | 0.272 | 0.238 | 0.174 |
| VILLAMA$_{QA}$ | 0.067 | 0.069 | 0.210 | **0.052** | 0.099 |
| VILLAMA | **0.061** | **0.059** | 0.139 | 0.140 | **0.099** |

Table 4: Results for each method for the ESS evaluation. The overall average NMSE for the four chapters. The best performance is indicated in **bold**, while the second best performance is indicated with underlined. Overall, VILLAMA (ours) achieves the best performance in predicting specific issues based on the group value distribution.

task targeting everyday life scenarios, this task concerns various issues, such as political, social, and religious ones.

**Setup**  In this experiment, we utilized a subset of the ESS, excluding the PVQ. ESS consists of each respondent's demographic information, such as gender, age, and family relationships, and survey questions in various topics, such as Understanding Democracy, Digital Social Contacts, and Attitudes to Climate Change. We first excluded questions in ESS that are not common across the participating countries. Then, we asked LLMs to answer the questionnaires in the following chapters in ESS:

- **Media and Social Trust (MST):** Media interest, beliefs and relationships with members of society, 5 questions.

- **Personal and Social Well-Being (PSWB):** Personal emotions and life satisfaction such as depression, happiness, and achievement, 39 questions.

- **Politics (POL):** Government, belief in the political system, opinions on immigrants, 34 questions.

- **Understanding of Democracy (UD):** Stance on various issues in the democratic system, 45 questions.

We created prompts for this task using the template in Table 21. We evaluated a given LLM's ability to predict opinions on specific issues by comparing its responses to the actual responses by the group whose value distribution was targeted.

Note that ESS questions use diverse response scales, including binary responses (0 or 1) and degrees of agreement (0 to 10). We rescaled response scores to the range of 0 to 1 to prevent certain questions from having a greater impact on the NMSE.

**Results**  Table 4 shows the results for the opinion prediction task. Overall, VILLAMA generates answers that align with the target value distribution more effectively than the other LLMs; VILLAMA achieves the best or second-best performance for four chapters, and it also exhibits the lowest average. These results suggest that VILLAMA can predict human opinions on specific issues more accurately based on the value distribution. Also, LLAMA$_{Short}$ exihibits a noticeably worse performance than LLAMA$_{Long}$, indicating that including value definitions in the prompt has a significant impact on the outcome. In the paired t-test results of LLAMA$_{Long}$ and VILLAMA, VILLAMA's improvement over LLAMA$_{Long}$ is statistically significant ($p < 0.001$) in MST, PSWB, and POL. In addition, we observe the tendency of ChatGPT$_{Long}$ to avoid answering questions related to opinion prediction.[3] Further analysis of ChatGPT's tendency to refuse to answer is in Appendix E.3. Example results of this task are provided in Table 27.

---

[3]The response starts with "I cannot answer this question as it goes against the ethical guidelines of OpenAI."

## 7 Conclusion

In this paper, we introduced the Value Injection Method (VIM), which allows for the injection of specific value distributions into existing LLMs through argument generation and question answering tasks. To assess the effectiveness of VIM across various value distributions, we conducted evaluations on 28 country groups and 100 social groups. The evaluations involved answering value surveys and generating arguments based on the given value distribution. Our results demonstrate that VIM outperforms other prompting methods in these evaluations. Additionally, we examined the efficacy of value injection and its ability to predict human behavior through behavior prediction and opinion prediction tasks. The empirical experiments conducted on these evaluation tasks confirm the effectiveness of VIM in value injection and its superior performance compared to other prompting methods in predicting human behaviors.

## Limitations

For VILLAMA$_{AG}$, we set a fixed hyper-parameter $\gamma$, which serves as the threshold for selecting the likelihood of the answer as three. The chosen number is intuitive, considering that the score range of Schwartz values is from one to six. However, appropriate $\gamma$ value may vary depending on the specific value distribution. While VIM demonstrates superior performance compared to other baselines in various value-related tasks, further improvements could be achieved by exploring the effectiveness of different values for $\gamma$.

The LLM trained by VIM has the ability to generate personalized answers based on an individual's value distribution. However, our exploration has been limited to group value distributions due to the lack of individual-level Schwartz value datasets. In the future, we will collect individual-level Schwartz value distribution data and examine the distinctions between the individual and group levels.

## Ethics Statement

VIM has the ability to simulate the behaviors and opinions of a group by injecting a specific value distribution into the LLM. However, one ethical concern is the potential misuse of VIM to imitate the stance or behavior of specific individuals without their explicit consent. Let us assume that if one possesses an individual's Schwartz value distribution information, it becomes possible that LLM with VIM can generate sentences that were not actually spoken from them. This raises concerns, especially for celebrities or public figures who share extensive personal information, as it may make them more susceptible to vulnerabilities such as the dissemination of fake news through misuse. To address this issue, employing a discriminator that can distinguish between speech generated by an LLM trained on values through VIM and authentic speech of individuals could be considered as a preventive measure.

In our human evaluation process, we ensure that annotators are compensated more than the minimum wage.

## Acknowledgements

We would like to thank the anonymous reviewers for their helpful questions and comments. This project is partially supported by Microsoft Research Asia. This work was partly supported by Institute of Information & communications Technology Planning & Evaluation (IITP) grant funded by the Korea government (MSIT) (No.2022-0-00680, Abductive inference framework using omni-data for understanding complex causal relations & ICT Creative Consilience program (IITP-2023-2020-0-018)). And this work was partially supported by the New Faculty Startup Fund from Seoul National University.

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

# Appendix

## A The number of Clusters

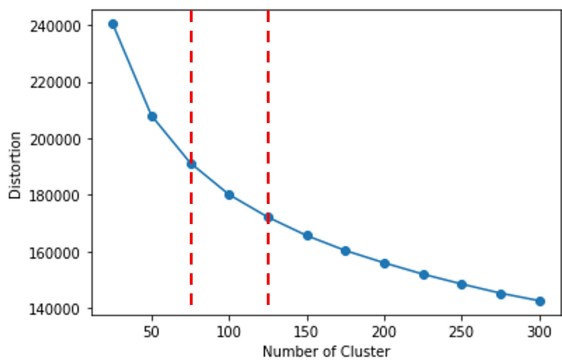

Figure 4: Elbow method graph. At the point where the number of clusters reaches 100, it is observable that the graph exhibits a curvature.

To determine the appropriate number of clusters, we employ the elbow method (Thorndike, 1953). Figure 4 presents the results of this analysis. We observe a curvature in the graph when the number of clusters reaches 100, indicating a potential elbow point. So we set the number of social groups to 100.

## B Implementation Details

We train LLAMA-7B (Touvron et al., 2023) which has a parameter size of seven billion using Pytorch on an NVIDIA RTX A6000 GPU, with 48GB dedicated memory. We use AdamW optimizer (Loshchilov and Hutter, 2019), train 5 epochs for fine-tuning, set batch size as 4, learning rate as 2e-5. We set the rank of LoRA, using the decomposition matrix to 8 and set the $\gamma$ of Argument

Generation of VIM to 3 which is the middle of the range of the values. In the inference process, we set temperature as 1 and top-p as 0.5. We use May 24, 2023 version of ChatGPT.[4]

## C Few-shot Results

For the opinion prediction (ESS) task among the evaluation tasks, we conducted experiments not only for zero-shot but also for additional few-shot prompting and few-shot prompting using the Chain of Thought (CoT). We experimented with 1, 2, and 5 examples, as the input context window of the LLM limited us from conducting experiments with a larger number of examples. Few-shot examples were randomly selected from the ESS dataset, and the prompts were carried out in the same manner as the zero-shot setting. The results are presented in the following table 5 and 6.

For value survey (PVQ) task, we conducted experiments with the version known for having a larger number of questions and being more accurate, which consists of 40 questions. However, the dataset we have is based on a 21-question version, which has a lower number of questions and lower accuracy. In this case, since we would have had to arbitrary assign answers to the remaining questions, we were unable to conduct few-shot experiments. The Behavior prediction task is answering with "agree" or "disagree" regarding whether the model would engage in the same action as a specific value-related scenario. Our evaluation focuses not on individual answers, but on assessing the percentage of "agree" or "disagree". Similar to Value survey task, there is a challenge of assigning arbitrary answers for few-shot examples, so we were unable to conduct few-shot experiments.

We found that VILLAMA with VIM applied achieved a significantly lower average normalized mean squared error (NMSE) of 0.099 compared to both few-shot and few-shot CoT settings. In the few-shot experiments, both LLAMA$_{Long}$ and ChatGPT$_{Long}$ showed their best performance in the zero-shot setting. In the few-shot CoT experiments, LLAMA$_{Long}$ showed the best performance with 5-shot, while ChatGPT$_{Long}$ performed best with 1-shot.

---

[4]https://help.openai.com/en/articles/6825453-chatgpt-release-notes

| Model | Example | MST | PSWB | POL | UD | Ave. |
|---|---|---|---|---|---|---|
| LLAMA_Long | 0-shot | 0.079 | 0.115 | 0.281 | **0.072** | 0.137 |
| | 1-shot | 0.154 | 0.109 | 0.533 | 0.250 | 0.261 |
| | 2-shot | 0.118 | 0.117 | 0.271 | 0.334 | 0.210 |
| | 5-shot | 0.165 | 0.109 | 0.209 | 0.103 | 0.147 |
| ChatGPT_Long | 0-shot | 0.222 | 0.133 | **0.106** | 0.216 | 0.169 |
| | 1-shot | 0.211 | 0.265 | 0.257 | 0.237 | 0.243 |
| | 2-shot | 0.174 | 0.275 | 0.280 | 0.263 | 0.248 |
| | 5-shot | 0.174 | 0.299 | 0.284 | 0.282 | 0.259 |
| VILLAMA | 0-shot | **0.061** | **0.059** | 0.139 | 0.140 | **0.099** |

Table 5: Results of few-shot experiments for the ESS evaluation. The overall average NMSE for the four chapters. The best performance is indicated in **bold**, while the second best performance is indicated with underlined. Overall, VILLAMA(ours) achieves the best performance in predicting specific issues based on the group value distribution.

| Model | Example | MST | PSWB | POL | UD | Ave. |
|---|---|---|---|---|---|---|
| LLAMA_Long | 0-shot | 0.066 | **0.040** | 0.573 | **0.056** | 0.184 |
| | 1-shot | 0.122 | 0.069 | 0.769 | 0.066 | 0.257 |
| | 2-shot | 0.103 | 0.069 | 0.432 | 0.109 | 0.178 |
| | 5-shot | 0.195 | 0.092 | **0.084** | 0.161 | 0.133 |
| ChatGPT_Long | 0-shot | 0.611 | 0.301 | 0.302 | 0.198 | 0.353 |
| | 1-shot | 0.172 | 0.222 | 0.239 | 0.219 | 0.214 |
| | 2-shot | 0.668 | 0.249 | 0.238 | 0.230 | 0.346 |
| | 5-shot | 0.186 | 0.264 | 0.243 | 0.239 | 0.233 |
| VILLAMA | 0-shot | **0.061** | 0.059 | 0.139 | 0.140 | **0.099** |

Table 6: Results of few-shot **Chain-of-Thought (CoT)** experiments for the ESS evaluation. The overall average NMSE for the four chapters. The best performance is indicated in **bold**, while the second best performance is indicated with underlined. Overall, VILLAMA(ours) achieves the best performance in predicting specific issues based on the group value distribution.

## D Temperature & Top-p Adjustment

To investigate how temperature and top-p affect the NMSE of each of the three evaluation tasks: value survey, behavior prediction, and opinion prediction, we conduct the experiment in which we adjusted both temperature and top-p.

Table 7 presents the results of temperature adjustment. We conducted experiments by varying the temperature values to 0.2, 0.4, 0.6, 0.8, and 1.0 while keeping the top-p fixed at 0.5. The lowest NMSE was observed at 0.2, which corresponds to the lowest temperature for value survey tasks. However, for behavior and opinion predictions, the NMSE is lowest at the highest temperature of 1.0. Table 8 presents the results of adjusting the top-p parameter. We conducted experiments by varying the top-p values to 0.25, 0.50, and 0.75 while keeping the temperature fixed at 1.0. The best per-

| Temp. | Value Survey | Behavior Pred. | Opinion Pred. |
|---|---|---|---|
| 0.2 | **0.033** | 0.072 | 0.102 |
| 0.4 | 0.033 | 0.072 | 0.102 |
| 0.6 | 0.034 | 0.074 | 0.101 |
| 0.8 | 0.034 | 0.071 | 0.100 |
| 1.0 | 0.034 | **0.071** | **0.099** |

Table 7: Results of temperature adjustment: The NMSE for the Value survey, behavior prediction, and opinion prediction tasks are calculated for temperatures of 0.2, 0.4, 0.6, 0.8, and 1.0 when top-p is fixed at 0.5. The best performance is indicated in **bold**.

| Top-p | Value Survey | Behavior Pred. | Opinion Pred. |
|---|---|---|---|
| 0.25 | **0.033** | **0.071** | 0.100 |
| 0.50 | 0.034 | 0.071 | **0.099** |
| 0.75 | 0.033 | 0.073 | 0.101 |

Table 8: Results of top-p adjustment: The NMSE for the Value survey, behavior prediction, and opinion prediction tasks are calculated for top-p of 0.25, 0.50, and 0.75 when temperature is fixed at 1.0. The best performance is indicated in **bold**.

formance was observed in the value survey and behavior prediction tasks at the lowest top-p value of 0.25, while the opinion prediction task achieved the highest performance at a top-p value of 0.50. However, both results indicated that when temperature and top-p were adjusted, the difference in NMSE was less than 0.005, suggesting that these adjustments did not have a significant impact on the results.

## E Additional Analyses

This section describes the additional analyses conducted throughout the experiment and evaluation process.

### E.1 Results of Evaluation 2: Argument Generation

In Figure 3, which is the result of Evaluation 2: Argument Generation, we proceeded to conduct an additional analysis addressing the following questions.

First, *why does* VILLAMA_QA *worse than baseline?* This is because of the prompts used for AG method in VIM are constructed with only two possibilities: whether they are "would say the {argument}" or "would not say the {argument}" to the target value distribution. This approach can be challenging to learn the value distribution properly. On the other hand, in the case of the QA method, it is

| "Assisted suicide should be a criminal offense" | |
| --- | --- |
| Model | Argument (Generated) |
| LLAMA$_{Long}$ | I agree. I think that it is not a good idea to kill someone. |
| VILLAMA$_{QA}$ | I agree. Assisted suicide should be a criminal offence because it is a form of murder. |
| VILLAMA | I agree. Assisted suicide should be a criminal offence because it is against the law to take another person's life. |

Table 9: Argument generated by LLAMA$_{Long}$, VILLAMA$_{QA}$, and VILLAMA for topic "Assisted suicide should be a criminal offense". The value scores of the target value distribution used in the generation are as follows: achievement=2.3, benevolence=2.1, conformity=2.9, hedonism=2.7, power=2.8, security=3.1, self-direction=2.1, stimulation=3.4, tradition=4.2, universalism=2.7.

| "We should legalize sex selection" | |
| --- | --- |
| Model | Argument (Generated) |
| LLAMA$_{Long}$ | I agree. It is a good way to prevent the gender imbalance. |
| VILLAMA$_{QA}$ | I agree. It is a natural process and it is not harmful to anyone. |
| VILLAMA | I agree. Sex selection is a natural right of every individual. |

Table 10: Argument generated by LLAMA$_{Long}$, VILLAMA$_{QA}$, and VILLAMA for topic "We should legalize sex selection". The value scores of the target value distribution used in the generation are as follows: achievement=4.2, benevolence=1.5, conformity=3.9, hedonism=2.2, power=5.0, security=1.7, self-direction=2.0, stimulation=3.9, tradition=1.8, universalism=1.6.

learned with six appropriate answers corresponding to different values. As a result, it can be considered that it learns the value distribution relatively better.

Second, *why does* VILLAMA *have a higher lose ratio compared to* VILLAMA$_{QA}$? VILLAMA$_{QA}$ has a relatively high tie ratio. This is because the VILLAMA$_{QA}$ generates arguments similar to the baseline. These are two argument generation examples for the topic "Assisted suicide should be a criminal offense" and "We should legalize sex selection" using LLAMA$_{Long}$, VILLAMA$_{QA}$, and VILLAMA.

Table 9 shows the generated arguments of the target value distribution. "Tradition" score is the highest at 4.2. The arguments generated by the LLAMA$_{Long}$ and VILLAMA$_{QA}$ have a looser connection with tradition and can be interpreted as aligning with other values such as benevolence or universalism. When comparing the LLAMA$_{Long}$ and VILLAMA$_{QA}$ in the human evaluation process of selecting arguments for groups with the target distribution, it becomes challenging to make decisions. Therefore, when comparing LLAMA$_{Long}$ and the VILLAMA$_{QA}$, due to the presence of similar arguments, the tie ratio can increase, leading to a relatively lower lose ratio as a result. On the other hand, the argument generated by the full VILLAMA shows a clear relationship to tradition, such as the basis of "laws should not be violated". VILLAMA effectively captures the characteristics of the target value distribution, generating arguments closely related to specific values. Due to these instances, the win ratio is higher for VILLAMA compared to when using only AG or QA, where it successfully identifies the context of specific values and generates relevant arguments.

In the case of VILLAMA , there are situations where it fails to consider other values within the value distribution when generating arguments associated with specific values. Table 10 is an example of such a case and the target value distribution. For the topic "We should legalize sex selection," VILLAMA generated an argument associated with the value "Stimulation" which shows a high score of 3.9 within the value distribution. However, the LLAMA$_{Long}$ also generated an argument related to the value "Power" which scored 5.0, another high-scoring value. However, as described above, VILLAMA$_{QA}$ often generates relatively similar arguments to LLAMA$_{Long}$, so this difference is small, which can be considered to have a low loss ratio in QA and a relatively large loss ratio in VILLAMA.

## E.2 Cluster Size and NMSE

We examined how each cluster size, which corresponds to the target value distribution, influences PVQ, behavior and opinion prediction tasks. The relationship between the cluster size and the NMSE for each task is illustrated in Figure 5. The NMSE

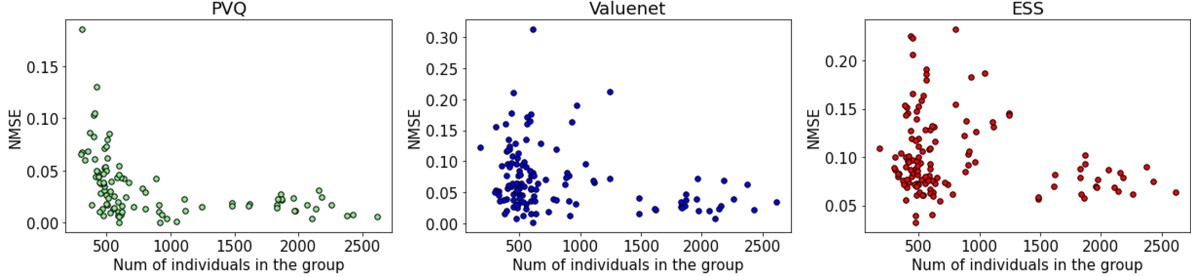

Figure 5: The NMSE results for each cluster in the PVQ, behavior prediction (Valuenet), and opinion prediction (ESS) tasks are presented. The cluster sizes range from a minimum of 511 to a maximum of 2616.

|  | MST | PSWB | POL | UD | Ave. |
|---|---|---|---|---|---|
| NMSE | 0.222 | 0.133 | 0.106 | 0.216 | 0.169 |
| Avoidance (%) | 12.2 | 28.1 | 29.7 | 0.6 | 17.7 |

Table 11: The NMSE and avoidance response ratio in the opinion prediction task of ChatGPT.

is commonly observed to be low in clusters with relatively large sizes, but in clusters with small sizes, it is sometimes measured to be high. This phenomenon appears to be attributed to the fact that larger groups tend to exhibit a more pronounced common value distribution, lifestyle, or opinion, while in smaller groups, the influence of a single member becomes more significant.

### E.3 ChatGPT Response Avoidance Ratio in Opinion Prediction

In the opinion prediction task, we observed that ChatGPT sometimes responds with 'I can't answer the questions because I'm an AI language model.' Since NMSE was calculated excluding these responses, we investigated the extent of response avoidance and its impact. Table 11 shows the NMSE and response avoidance ratio of ChatGPT in the opinion prediction task.

ChatGPT exhibited the highest response avoidance ratio in the 'Politics' of the ESS, at 29.7%, and the lowest avoidance ratio in 'Understanding Democracy,' at 0.6%. These findings confirm that high avoidance ratios contribute to the observed low NMSE.

### F Dataset Examples

This section presents the examples of datasets. We use four datasets in this paper as follows:

- Touché23-ValueEval - Table 12
- VALUENET - Table 13
- Portrait Values Questionnaire - Table 14
- European Social Survey - Table 15

### G Prompts

This section describes the prompts used to train LLMs by VIM. The prompts are as follows:

- VIM Argument Generation prompt - Table 16
- VIM Question Answering prompt - Table 17

Prompts for four different tasks are provided:

- PVQ task prompt - Table 18
- Argument Generation task prompt - Table 19
- VALUENET task prompt - Table 20
- ESS task prompt - Table 21

Furthermore, Table 22 presents the basic prompt used to provide the target group with Schwartz value distribution, while Table 23 shows the in-context learning prompt, which includes the definition of Schwartz values and the value distribution.

### H Experiments Result Examples

This section describes the examples of experiment results. The examples are presented as follows:

- PVQ task results - Table 24
- Argument Generation task results - Table 25
- VALUENET task results - Table 26
- ESS task results - Table 27

### I Human Evaluation

This section presents the human evaluation conducted for the argument generation task. Since there is no ground-truth argument based on the value distribution, we utilize Google Form for the evaluation. A screenshot of the questionnaire can be seen in Figure 6.

| Conclusion | Stance | Premise | Value |
|---|---|---|---|
| We should abandon the use of school uniform | In favor of | School uniforms are too expensive and many parents can not afford the extra expense. | Security |
| We should abolish the Olympic Games | Against | Olympic Games should not be banned. They are the tissue that solidify the friendship between nations in the world and a reminder of a peaceful world which should be the world vision. | Benevolence, Universalism |
| We should adopt gender-neutral language | In favor of | We should adopt gender-neutral language because it avoids offending people with gender stereo-types. | Conformity, Universalism |

Table 12: Examples of Touché23-ValueEval Dataset. Conclusion is specific topic, stance refers to the agreement or disagreement with the conclusion. The premise represents the reasons for the stance on the conclusion, and value encompasses all the values expressed in the premise.

| Value | Stance | Scenario |
|---|---|---|
| Achievement | Negative | I am the original loser. |
| Tradition | Positive | Not wanting my kids to eat candy for breakfast |
| Universalism | Negative | I hate seeing posts about people who have their lives together because I don't |

Table 13: Examples of VALUENET Dataset. Value represents one of the Schwartz values, while stance indicates whether the scenario is positive or negative to the value. The scenario depicts everyday actions associated with the value.

| No | Items |
|---|---|
| 1 | Thinking up new ideas and being creative is important to him. He likes to do things in his own original way. |
| 2 | He seeks every chance he can to have fun. It is important to him to do things that give him pleasure. |
| 3 | He strongly believes that people should care for nature. Looking after the environment is important to him. |

Table 14: Examples of Portrait Values Questionnaire Item. The items depict individuals involved in actions associated with the respective value.

| No | Items |
|---|---|
| 1 | Please answer 0 to 10 where 0 is not at all important for democracy in general and 10 is extremely important for democracy in general. |
| 2 | Is your group discriminated against by other grounds? Please answer "Yes" or "No". |
| 3 | What extent do you think those who hold extreme political views in your country today are prevented from expressing them openly? Please answer 0 to 10 where 0 is not at all and 10 is completely. |

Table 15: Examples of European Social Survey Item. The item is comprised of inquiries pertaining to political and social issues.

| VIM | Prompt |
|---|---|
| Argument Generation | Tell me what you would say about the following statement. Statement: {conclusion} Answer: 

 **Answer type 1** 
 I would say, "I {stance} with that because {premise}." about {conclusion} 

 **Answer type 2** 
 I would not say, "I {stance} with that because {premise}." about {conclusion} |

Table 16: VIM: Argument Generation Prompt. { } represents the placeholder. {conclusion}, {stance} and {premise} in the placeholder are element of the Touché23-ValueEval dataset. This prompt pertains to determining what answer to generate for the conclusion. Answer type 1 signifies that the target group would express the argument, while answer type 2 indicates that the target group would not endorse or articulate the same argument. The choice between generating Answer type 1 or Answer type 2 is based on $w_a$ in Algorithm 1.

| VIM | Prompt |
|---|---|
| Question
Answering | **Prompt**
Indicate for the following statement whether it is
1. 'Not like me at all',
2. 'Not like me',
3. 'A little like me',
4. 'Somewhat like me',
5. 'Like me',
6. 'Very much like me'
as a description of you.
Statement: I {stance} with {conclusion}. {premise}.
Answer:

**Answer**
Because I think {value} is {EXPRESSION}, my answer is {score}. |

Table 17: VIM: Question Answering Prompt. { } represents the placeholder. {conclusion}, {stance} and {premise} in the placeholder are element of the Touché23-ValueEval dataset. Examples of Touché23-ValueEval dataset are table 12. Value is one of the Schwartz values, and EXPRESSION is a textual representation that aligns with a numeric scale ranging from 1 to 6. (1: not important to me at all, 2: not important to me, 3: a little important to me, 4: somewhat important to me, 5: important to me, 6: very much important to me). Prompt inquires about the degree of similarity between the provided statement and one's own self, while the answer denotes the appropriate response based on a particular value.

| Task | Prompt |
|---|---|
| PVQ | **Prompt**
Indicate for the following statement whether it is
1. 'Not like me at all',
2. 'Not like me',
3. 'A little like me',
4. 'Somewhat like me',
5. 'Like me',
6. 'Very much like me'
as a description of you.
Statement: {PVQ item}.
Answer: |

Table 18: Task 1 - Portrait Values Questionnaire Prompt. { } represents the placeholder. {PVQ item} in the placeholder are element of the PVQ. Prompt asks how similar the given statement is to oneself, while the answer is one of the options in prompt. Examples of PVQ item are table 14 and generation results are table 24.

| Task | Prompt |
|---|---|
| Argument
Generation | **Prompt 1 - stance**
Do you agree or disagree with {conclusion}?
Answer:

**Prompt 2 - premise**
Explain in detail why you {stance} with {conclusion}.
Answer: |

Table 19: Task 2 - Argument Generation Prompt. { } represents the placeholder. {conclusion}, {stance} and {premise} in the placeholder are element of the Touché23-ValueEval dataset. Prompt 1 serves the purpose of generating a stance, while Prompt 2 serves the purpose of generating a premise. Examples of Touché23-ValueEval dataset are table 12 and generation results are table 25.

| Task | Prompt |
|---|---|
| VALUENET | **Prompt**
Do you agree the behavior or expression in the following scenario similar to you?
Please answer "I agree" or "I disagree".
Scenario: {VALUENET scenario}.
Answer: |

Table 20: Task 3 - VALUENET Prompt. { } represents the placeholder. {VALUENET scenario} is the actions of people related to values in everyday life. The purpose of this prompt is to generate responses that agree or disagree with whether one would engage in similar actions or expressions to the given scenario. Examples of VALUENET dataset are table 13 and generation results are table 26.

| Task | Prompt |
|------|--------|
| ESS | {ESS item}. |
|      | Answer: |

Table 21: Task 4 - European Social Survey (ESS) Prompt Example. { } represents the placeholder. {ESS item} means the questions of the European Social Survey. Prompt asks the ESS item, while the answer is one of the options in prompt or digit. Examples of ESS item are table 15 and generation results are table 27.

| Short Prompt | Prompt |
|--------------|--------|
|              | Let's roleplay. |
|              | |
|              | Value Score: |
|              | - Achievement: {target Achievement score} |
|              | - Benevolence: {target Benevolence score} |
|              | - Conformity: {target Conformity score} |
|              | - Hedonism: {target Hedonism score} |
|              | - Power: {target Power score} |
|              | - Security: {target Security score} |
|              | - Self-Direction: {target Self-Direction score} |
|              | - Stimulation: {target Stimulation score} |
|              | - Tradition: {target Tradition score} |
|              | - Universalism: {target Universalism score} |
|              | As this person, please answer the following question. |
|              | You just have to choose the answer, you don't have to explain it. |
|              | Please choose only one option, even if you're not sure. |
|              | |
|              | {Task Description} |

Table 22: Short prompt for providing the target value distribution to LLM. { } represents the placeholder, and {target value score} means corresponding value score of target value distribution. {Task Description} the task description encompasses individual prompts designated for each of the four tasks delineated in Section 4, which are utilized as input for the respective tasks.

| Long Prompt | Prompt |
|---|---|
| | Let's roleplay.

I will describe a person who have values between 1 and 6 for each value.
1 means the value is not important to him at all and 6 means the value is very much important to him.

Value Definition:
- Achievement: values personal success through demonstrating competence according to social standards
- Benevolence: values preserving and enhancing the welfare of those with whom one is in frequent personal contact (the 'in-group')
- Conformity: values restraint of actions, inclinations, and impulses likely to upset or harm others and violate social expectations or norms
- Hedonism: values pleasure or sensuous gratification for oneself
- Power: values social status and prestige, control or dominance over people and resources
- Security: values safety, harmony, and stability of society, of relationships, and of self
- Self-direction: values independent thought and action–choosing, creating, exploring.
- Stimulation: values excitement, novelty, and challenge in life
- Tradition: values respect, commitment, and acceptance of the customs and ideas that one's culture or religion provides
- Universalism: values understanding, appreciation, tolerance, and protection for the welfare of all people and for nature

Value Score:
- Achievement: {target Achievement score}
- Benevolence: {target Benevolence score}
- Conformity: {target Conformity score}
- Hedonism: {target Hedonism score}
- Power: {target Power score}
- Security: {target Security score}
- Self-Direction: {target Self-Direction score}
- Stimulation: {target Stimulation score}
- Tradition: {target Tradition score}
- Universalism: {target Universalism score}
As this person, please answer the following question.
You just have to choose the answer, you don't have to explain it.
Please choose only one option, even if you're not sure.

{Task Description} |

Table 23: Long prompt for providing the target value distribution to LLM. { } represents the placeholder, and the variables to be inserted into the placeholders are the same as those in table 22. However, by providing additional input in the form of definitions of Schwartz values, we aim to elicit responses from the language model that are more similar to those of the target group, leveraging the enriched information.

| PVQ Item | Answer (Generated) |
|---|---|
| ViLLaMA | |
| Thinking up new ideas and being creative is important to him. He likes to do things in his own original way. | 3 |
| He seeks every chance he can to have fun. It is important to him to do things that give him pleasure. | 4 |
| He thinks it is best to do things in traditional ways. It is important to him to keep up the customs he has learned. | 2 |
| LLAMA$_{Long}$ | |
| Thinking up new ideas and being creative is important to him. He likes to do things in his own original way. | 5 |
| He seeks every chance he can to have fun. It is important to him to do things that give him pleasure. | 5 |
| He thinks it is best to do things in traditional ways. It is important to him to keep up the customs he has learned. | 4 |

Table 24: Example results of task 1 - PVQ. The results reflect the responses of ViLLaMAand LLAMA$_{Long}$. Score is ranging from 1 to 6, to indicate the extent of resemblance between LLM and the items presented.

| Conclusion | Premise (Genenrated) | Stance (Genenrated) |
|---|---|---|
| VɪLLaMA | | |
| We should abandon the use of school uniform | School uniforms are a good way to make students feel equal | Against |
| Assisted suicide should be a criminal offence | Assisted suicide should be a criminal offence because it is against the law to take another person's life | In favor of |
| We should ban human cloning | I think that human cloning is a good thing because it can help people who are in need of organs. | Against |
| LLaMA_{Long} | | |
| We should abandon the use of school uniform | It is not necessary to wear uniform to go to school | In favor of |
| Assisted suicide should be a criminal offence | I think that it is not a good idea to kill someone | In favor of |
| We should ban human cloning | I think that human cloning is a very bad thing because it is against the law. | In favor of |

Table 25: Example results of task 2 - Argument Generation. Applied the VIM and prompting methods to the value distribution from figure 2 to generate reasons and stances for the given topics. Two samples of each result example using VIM and the prompting method. VɪLLaMAgenerates arguments that values custom and law in relation to tradition rather than prompting.

| Scenario | Value | Stance | Answer (Generated) |
|---|---|---|---|
| VɪLLaMA | | | |
| I decided to make a budget so that I wouldn't go broke again | Achievement | Positive | Agree |
| Asking someone to respect me | Conformity | Positive | Agree |
| Going on a trip without my best friend, even though we planned it together | Hedonism | Negative | Disagree |
| LLaMA_{Long} | | | |
| I decided to make a budget so that I wouldn't go broke again | Achievement | Positive | Disagree |
| Asking someone to respect me | Conformity | Positive | Agree |
| Going on a trip without my best friend, even though we planned it together | Hedonism | Negative | Agree |

Table 26: Example results of task 3 - VALUENET. The generated responses aim to determine whether LLM agree or disagree with engaging in behaviors similar to scenarios that have a relationship with a specific value and stance.

| Question | Answer (Generated) |
|---|---|
| **VɪLLᴀMA** | |
| How important is it for you to live in a country that is governed democratically? Please tell me on a score of 0 to 10 where 0 is not at all important and 10 is extremely important. | 9 |
| Please tell me how important you think it is for democracy in general that national elections are free and fair? Please answer 0 to 10 where 0 is not at all important for democracy in general and 10 is extremely important for democracy in general. | 8 |
| Would you say it is generally bad or good for your country's economy that people come to live here from other countries? Please answer 0 to 10, where 0 means bad for the economy and 10 means good for the economy. | 5 |
| **LLᴀMA**$_{Long}$ | |
| How important is it for you to live in a country that is governed democratically? Please tell me on a score of 0 to 10 where 0 is not at all important and 10 is extremely important. | 5 |
| Please tell me how important you think it is for democracy in general that national elections are free and fair? Please answer 0 to 10 where 0 is not at all important for democracy in general and 10 is extremely important for democracy in general. | 7 |
| Would you say it is generally bad or good for your country's economy that people come to live here from other countries? Please answer 0 to 10, where 0 means bad for the economy and 10 means good for the economy. | 8 |

Table 27: Example results of task 4 - ESS. Through each method, LLM generate responses corresponding to the given question.

**[Group_60] Value Distribution**

Achievement : 2.2
Benevolence : 1.6
Conformity : 2.7
Hedonism : 2.0
Power : 4.0
Security : 2.0
Self-direction : 1.7
Stimulation : 1.8
Tradition : 3.7
Universalism : 1.7

Q1. We should stop the development of autonomous cars *

○ Agree, Autonomous cars will make people lazy and not care about the environment.

○ Agree, Self-driving cars will make people lazy and unhealthy because they will not exercise.

○ I don't know (tie)

Q2. Assisted suicide should be a criminal offence *

○ Agree, I think that it is not a good idea to kill someone.

○ Agree, Assisted suicide is a violation of the right to life, which is a fundamental right, as stated in the Universal Declaration of Human Rights.

○ I don't know (tie)

Figure 6: Example of Human Evaluation Question Form. Annotators solve the questions presented in the question box located at the bottom by examining the value distribution of the group positioned at the top. In regard to the conclusion stated in the question, they select the stance and premise among the two options that are most similar to the group possessing the corresponding value distribution.