# OpenReview forum: "From Values to Opinions: Predicting Human Behaviors and Stances Using Value-Injected Large Language Models"
_EMNLP/2023/Conference — EMNLP 2023 Main_

### Official Review · Reviewer_uZ69 · 2023-08-03

**Soundness:** 5

**Ethical Concerns:**

Yes

**Excitement:**

4: Strong: This paper deepens the understanding of some phenomenon or lowers the barriers to an existing research direction.

**Justification For Ethical Concerns:**

I think there might be ethical concerns with injecting human values into LLMs. There is some discussion in the paper on that, in the ethics statement section, but its not very thorough.

**Paper Topic And Main Contributions:**

The paper presents a method for Value Injection of human values into LLMs. The method uses data from the Touché23-ValueEval dataset of arguments, where each argument has a conclusion, stance, premise and values. The values are represented with a 10 dimensional binary vector, specifying whether the argument reflects or not one of the 10 human values in Schwartz values identify theory. The theory suggests that a human personality may be characterized by these 10 values.

The data is used to fine-tune a LAMMA model, by formulating argument generation and question answering tasks posed using the dataset.

The value-injected model ViLLaMA, is then evaluated in multiple experiments. The used baselines are variants of LLaMA without any fine-tuning, and variants trained only on argument generation or questions answering. Also included is a comparison with ChatGPT.

The presented experiments explore (1) the personality of the LLM, evaluated by taking the PVQ questionnaire, (2) the ability to generate arguments matching a given personality, (3) the ability to predict the behavior of people of a certain personality, and (4) whether its responses to questions about various issue (e.g. political issues) match the answers of a person with the same personality?

Results for the experiments show that ViLLaMA is typically the top-performer on the above tasks, from the considered baselines.

**Questions For The Authors:**

algorithm 5, line 5: suggestion - add in line 2 that V_g={v_g^z}, so the connection is clearer. It wasn't clear to me, at first.

algorithm 2, line 4: can this be clarified? The text says "we discretize the real numbers into integer l_a by considering the number of decimal places", and that does not seem to fit all the parts of this assignment?


**Reasons To Accept:**

-- The paper is well-written and clear

-- The explored topic is interesting and timely

-- The experimental setting is sound, and the explored questions are interesting

-- Results are convincing


**Reasons To Reject:**

-- None

**Reproducibility:**

4: Could mostly reproduce the results, but there may be some variation because of sample variance or minor variations in their interpretation of the protocol or method.

**Reviewer Confidence:**

4: Quite sure. I tried to check the important points carefully. It's unlikely, though conceivable, that I missed something that should affect my ratings.

**Typos Grammar Style And Presentation Improvements:**

typos/grammar

line 57: and investigated methods simulate human behaviors -> .. that simulate ... ?

line 77: resulting three value-injected -> missing word ?

line 86: to prediction opinions and behaviors -> to predict

line 285: (refer to Appendix A for more details -> add )

---

> ### Author Rebuttal · Authors · 2023-08-29
>
> Dear reviewer,
>
> We would like to appreciate you for taking the time to review our research. Thank you for all your constructive comments.
>
> Our responses to the questions are as follows:
> >  algorithm 5, line 5: suggestion - add in line 2 that V_g={v_g^z}, so the connection is clearer. It wasn't clear to me, at first.
>
> Thank you for your advice. We appreciate your suggestion to add V_g={v_g^z} in line 2, as it would provide better clarity in understanding our algorithm. We will describe the algorithm reflecting this change.
> - - -
> > algorithm 2, line 4: can this be clarified? The text says "we discretize the real numbers into integer l_a by considering the number of decimal places", and that does not seem to fit all the parts of this assignment?
>
> Thank you for pointing out this unclarity. We think the current explanation is a little bit hard to follow. We intended to use the Bernoulli distribution to prevent situations where rounding the value score always results in only one single score. If the value score is determined as a single score through rounding, there exists a problem where the score will always be lower or higher than the actual score, making it impossible to accurately represent it.
>
> e.g.)
>
>     Achievement score = 4.7
>     Score (using round): 5 (always)
>     Score (using Bernoulli distribution): 4 or 5 (4 = 30%, 5 = 70%)
>
> We will revise the sentence "we discretize the real numbers into integer l_a by considering the number of decimal places" to "If an average value score is not a whole number, we rounded it down or up to an integer probabilistically based on its fractional part (e.g., if the score is 5.2, then it is rounded to 5 with an 80% chance or to 6 with a 20% chance)." with an example added to it for a clearer expression. Furthermore, we will continue to carefully examine the algorithm and review it to enhance clarity, ease of understanding, and reproducibility.

---

### Official Review · Reviewer_1Yni · 2023-08-05

**Typos Grammar Style And Presentation Improvements:** 1. In line 285, missing right bracket.
**Soundness:** 3

**Excitement:**

4: Strong: This paper deepens the understanding of some phenomenon or lowers the barriers to an existing research direction.

**Missing References:**

None.

**Paper Topic And Main Contributions:**

This paper focuses on predicting human behaviors and stances. To tackle this problem, the author proposes a value-injected large language model which can inject human values with the help of argument generation and question answering via LORA fine-tuning. The experimental results show that the proposed methods can properly inject human values and outperformers vanilla large language models like LLaMA and ChatGPT. The main contributions are three folds: 1) the authors introduce a novel problem to predict human behaviors and opinions with specific values; 2) the authors present a Value Injection Method (VIM) for injecting human values into LLMs; 3) the detailed experiments prove the feasibility of VIM.

**Questions For The Authors:**

A. How do you set the hyper-parameters of each LLM, and how do the temperature and top-p influence the performance?

**Reasons To Accept:**

1. The idea of injecting human values into LLM is quite novel.
2. Detailed experimental analysis.


**Reasons To Reject:**

1. There is a problem with the expression of contribution 1. I deem that the novel problems should emphasize predicting human behaviors and opinions with specific values instead of using value-injected LLMs.
2. The experimental results shown in this paper need some significance tests to verify the VIM's effectiveness further.
3. Missing the hyper-parameters of LLMs.


**Reproducibility:**

4: Could mostly reproduce the results, but there may be some variation because of sample variance or minor variations in their interpretation of the protocol or method.

**Reviewer Confidence:**

3: Pretty sure, but there's a chance I missed something. Although I have a good feel for this area in general, I did not carefully check the paper's details, e.g., the math, experimental design, or novelty.

---

> ### Author Rebuttal · Authors · 2023-08-29
>
> Dear reviewer,
>
> We would like to appreciate you for taking the time to review our research. Thank you for all your constructive comments.
>
> Our responses to the questions are as follows:
> > How do you set the hyper-parameters of each LLM, and how do the temperature and top-p influence the performance?
>
> We submitted the experimental code for reproduction and specified the hyperparameters in the shared code. However, we missed including this information in the paper. The LLMs used in the experiments are LLaMA-7B and ChatGPT (gpt-3.5-turbo). For LLaMA-7B, we used a temperature of default value, 1.0 and top-p of 0.5. For ChatGPT, we utilized the OpenAI API with a temperature of the default value, 1.0, and no further additional settings were applied. We will add these detailed parameter settings to the Implementation Details section of Appendix E. And we hope this will help with reproducibility.
>
> During the experiments, due to resource and time constraints, we intuitively set parameters such as gamma, learning rate, and the rank of LoRA, in addition to the previously mentioned temperature and top-p. We will conduct additional experiments to explore how top-p and temperature affect the performance, and we will describe the results in the final paper.
> - - -
> > There is a problem with the expression of contribution 1. I deem that the novel problems should emphasize predicting human behaviors and opinions with specific values instead of using value-injected LLMs.
>
> Thank you for your advice. We believe that your suggestion clarifies our intended contribution even further. As advised, we will make the revision to the expression of contribution 1, changing it from "using value-injected LLMs" to "with specific values."
> - - -
> > The experimental results shown in this paper need some significance tests to verify the VIM's effectiveness further.
>
> Yes, we believe significance test is a useful way to demonstrate the effectiveness of VIM. To address the points you mentioned, we adopt a paired t-test that shows how much the method affects the results. We verified the effectiveness of VIM by comparing the results of LLaMA-7B and ViLLaMA on PVQ, Valuenet, and ESS tasks.
> For the PVQ and ESS tasks, ViLLaMA's improvement over LLaMA-7B is statistically significant (p-value < 0.001). For the Valuenet task, it varied by value; the improvement was statistically significant for Achievement and Self-direction (p-value <= 0.001), but no significance was found for the other values.
> Based on these results, we will continue our efforts to enhance the performance in these tasks as future work and describe these results and explanations in the final paper.

---

### Official Review · Reviewer_FesA · 2023-08-05

**Soundness:** 3

**Excitement:**

3: Ambivalent: It has merits (e.g., it reports state-of-the-art results, the idea is nice), but there are key weaknesses (e.g., it describes incremental work), and it can significantly benefit from another round of revision. However, I won't object to accepting it if my co-reviewers champion it.

**Paper Topic And Main Contributions:**

The paper proposed to use value-injected LLMs to predict opinions and behaviors of people. two methods: argument generation and question answering are designed to inject targeted value distributions into LLMs via fine-tuning.


**Reasons To Accept:**

(1) The idea is interesting.
(2) Extensive experiments have been done to prove its efficacy.
(3) Te paper is well written and easy to follow.



**Reasons To Reject:**

(1) The novelty of the paper is limited. The idea is mainly to leverage LLM through fine-tuning, which has been extensively studied before. No novel techniques are proposed.

**Reproducibility:**

3: Could reproduce the results with some difficulty. The settings of parameters are underspecified or subjectively determined; the training/evaluation data are not widely available.

**Reviewer Confidence:**

4: Quite sure. I tried to check the important points carefully. It's unlikely, though conceivable, that I missed something that should affect my ratings.

---

> ### Author Rebuttal · Authors · 2023-08-29
>
> Dear reviewer,
>
> We would like to appreciate you for taking the time to review our research. Thank you for all your constructive comments.
>
> Our responses to the questions are as follows:
> > The novelty of the paper is limited. The idea is mainly to leverage LLM through fine-tuning, which has been extensively studied before. No novel techniques are proposed.
>
> Yes, as you mentioned, fine-tuning is a well-established technique. However, how to leverage it to effectively inject the value distribution into a model is an unexplored area, and we believe that finding an effective method out of many possible approaches is also a significant technical contribution. Just like instruction tuning is essentially fine-tuning but significantly impacted the paradigm of language models by the way the training data are designed, we proposed the argument generation (AG) and question answering (QA) methods rather than simply fine-tuning on existing labeled data. In each approach, AG and QA, we provided working solutions for how to select arguments and transform them into appropriate training data. This enables suitable learning for all conceivable value distributions.
>
> Our main contribution to this paper is not only proposing the Value Injection Method (VIM) but also predicting human behaviors and opinions using LLMs. We tackled the question of whether it is possible to predict the human behavior and opinions of specific groups using LLM, and we have shown the feasibility of value-injected LLMs in predicting them without directly training on the individual tasks.
> Compared to traditional methods like surveys, this approach has the significant advantage of being very cost-effective. There are various possibilities for applications such as policy formulation, demand analysis, marketing, and more.
> - - -
> > Could reproduce the results with some difficulty. The settings of parameters are underspecified or subjectively determined; the training/evaluation data are not widely available
>
> We submitted the experimental code for reproduction and specified the hyperparameters in the shared code. However, we missed including this information in the paper. Thank you for bringing this to our attention. The LLMs used in the experiments are LLaMA-7B and ChatGPT (gpt-3.5-turbo). For LLaMA-7B, we used a temperature of default value, 1.0 and top-p of 0.5. For ChatGPT, we utilized the OpenAI API with a temperature of the default value, 1.0, and no further additional settings were applied. We will add these detailed parameter settings to the Implementation Details section of Appendix E. And we hope this will help with reproducibility.
>
> Furthermore, all datasets we used are publicly available and the reference of each dataset is already specified in our paper: Touche23-ValueEval, Valuenet, and the European Social Survey. Anyone interested can easily access these datasets and apply VIM using the described methodology. We will revise the writing to provide clearer instructions and additional explanations, making it even easier for others to follow the dataset and VIM procedure.

---

### Official Review · Reviewer_vyUS · 2023-08-06

**Soundness:** 4

**Excitement:**

4: Strong: This paper deepens the understanding of some phenomenon or lowers the barriers to an existing research direction.

**Paper Topic And Main Contributions:**

This paper proposes to use value-injected large language models (LLM) to predict opinions and behaviors. The author proposes Value Injection Method (VIM) which combines argument generation and question answering to inject targeted value distributions into LLMs via fine-tuning. Through a series of experiments, the authors found that the proposed models are significantly better than the baselines.

Overall I think it is a very interesting task and I can definitely see its value. I have several questions, comments and concerns.
1. In figure 3, it seems that VILLAMAag is even worse than the baseline model, moreover, the full VILLAMA actually loses more than the VILLAMAqa model. Any idea why? It can be helpful to provide more analysis on the labeling result and see in what cases the full VILLAMA model improves and in what cases become worse.
2. I am a bit concerned with the setting of the baseline. Since you are fine-tuning on the data, comparing with zeroshot prompting is not fair. fewshot prompting and fewshot CoT would be better baselines and are necessary.
3. More ethical discussions are needed. I can see the value of the task, however, it can also create societal harm as you are predicting public opinions and values. Marginalized groups values may not be well representative in the data and it would be good to see the distribution regarding different groups of people.

**Reasons To Accept:**

1. A super interesting task
2. The proposed method (at least VILLAMA qa) seems to be working

**Reasons To Reject:**

1. Missing comparisons with several key baselines (e.g. few-shot prompting and few-shot CoT prompting)
2. The proposed VILLAMA AG method does not seems to always work, more analysis on this is needed
3. More ethical discussions are needed as the task is sensitive and might cause social harm.

**Reproducibility:**

4: Could mostly reproduce the results, but there may be some variation because of sample variance or minor variations in their interpretation of the protocol or method.

**Reviewer Confidence:**

4: Quite sure. I tried to check the important points carefully. It's unlikely, though conceivable, that I missed something that should affect my ratings.

---

> ### Author Rebuttal · Authors · 2023-08-29
>
> Dear reviewer,
>
> We would like to appreciate you for taking the time to review our research. Thank you for all your constructive comments.
>
> Our responses to the questions are as follows:
> > In Figure 3, it seems that VILLAMAag is even worse than the baseline model, moreover, the full VILLAMA actually loses more than the VILLAMAqa model. Any idea why? It can be helpful to provide more analysis on the labeling result and see in what cases the full VILLAMA model improves and in what cases becomes worse.
>
> > First question, why is ViLLaMA AG worse than baseline?
>
> The prompts used for AG training are constructed with only two possibilities: whether they are “would say the {argument}” or “would not say the {argument}” to the target value distribution. This approach can be challenging to learn the value distribution properly.
> On the other hand, in the case of the QA method, it is learned with six appropriate answers corresponding to different values. As a result, it can be considered that it learns the value distribution relatively better.
>
> > Second question, why does Full ViLLaMA have a higher lose ratio compared to ViLLaMA QA?
>
> In Figure 3, ViLLaMA QA has a relatively high tie ratio. This is because the ViLLaMA QA generates arguments similar to the baseline. These are argument generation example results for the topic "Assisted suicide should be a criminal offense," using LLaMA (baseline), ViLLaMA QA, and ViLLaMA.
>
> This is the value distribution of target group.
> | Ach | Ben | Con | Hed | Pow | Sec |  SD | Sti | Tra | Uni |
> |:---:|:---:|:---:|:---:|:---:|:---:|:---:|:---:|:---:|:---:|
> | 2.3 | 2.1 | 2.9 | 2.7 | 2.8 | 3.1 | 2.1 | 3.4 | 4.2 | 2.7 |
>
> | Topic      | Assisted suicide should be a criminal offence                                                                        |
> |------------|----------------------------------------------------------------------------------------------------------------------|
> | Baseline    | I agree. I think that it is not a good idea to kill someone.                                                         |
> | ViLLaMA QA | I agree. Assisted suicide should be a criminal offence because it is a form of murder.                               |
> | ViLLaMA    | I agree.  Assisted suicide should be a criminal offence because it is against the law to take another person's life. |
>
> In the target distribution, "Tradition" score is the highest at 4.2.
> The arguments generated by the baseline and ViLLaMA QA have a looser connection with tradition and can be interpreted as aligning with other values such as benevolence or universalism.
> When comparing the baseline and ViLLaMA QA in the human evaluation process of selecting arguments for groups with the target distribution, it becomes challenging to make decisions. Therefore, when comparing baseline and the ViLLaMA QA, due to the presence of similar arguments, the tie ratio can increase, leading to a relatively lower lose ratio as a result.
>
> On the other hand, the argument generated by the full ViLLaMa shows a clear relationship to tradition, such as the basis of “laws should not be violated”.
> ViLLaMA effectively captures the characteristics of the target value distribution, generating arguments closely related to specific values. Due to these instances, the win ratio is higher for ViLLaMA compared to when using only AG or QA, where it successfully identifies the context of specific values and generates relevant arguments.
>
> However, in the case of ViLLaMA, there are situations where it fails to consider other values within the value distribution when generating arguments associated with specific values. The following is an example of such a case and value distribution of target group.
>
> | Ach | Ben | Con | Hed | Pow | Sec |  SD | Sti | Tra | Uni |
> |:---:|:---:|:---:|:---:|:---:|:---:|:---:|:---:|:---:|:---:|
> | 4.2 | 1.5 | 3.9 | 2.2 | 5.0 | 1.7 | 2.0 | 3.9 | 1.8 | 1.6 |
>
> | Topic      | We should legalize sex selection                                                                        |
> |------------|----------------------------------------------------------------------------------------------------------------------|
> | Baseline   | I agree. It is a good way to prevent the gender imbalance.                                                |
> | ViLLaMA QA | I agree. It is a natural process and it is not harmful to anyone.                               |
> | ViLLaMA    | I agree. Sex selection is a natural right of every individual. |
>
> For the topic "We should legalize sex selection," ViLLaMA generated an argument associated with the value "Stimulation" which shows a high score of 3.9 within the value distribution. However, the baseline also generated an argument related to the value "Power" which scored 5.0, another high-scoring value.
>
> However, as described above, ViLLaMA QA often generates relatively similar arguments to baseline, so this difference is small, which can be considered to have a low loss ratio in QA and a relatively large loss ratio in ViLLaMA.
>
> We view these limitations as part of our future work and will continue to make efforts to improve performance. Additionally, we will describe these examples and analysis through an extra page.
> - - -
> > I am a bit concerned with the setting of the baseline. Since you are fine-tuning on the data, comparing with zeroshot prompting is not fair. fewshot prompting and fewshot CoT would be better baselines and are necessary.
>
> Yes, we have taken into consideration the points you mentioned about the baseline. For the Opinion Prediction (ESS) task among the evaluation tasks, we conducted experiments not only for zero-shot but also for additional few-shot prompting and few-shot prompting using the Chain of Thought (CoT). We experimented with 1, 2, and 5 examples, as the input context window of the LLM limited us from conducting experiments with a larger number of examples. Few-shot examples were randomly selected from the ESS dataset, and the prompts were carried out in the same manner as the zero-shot setting. The results are presented in the following table.
>
> < Opinion Prediction (ESS) Few-Shot Results >
> |   Model  | Number of example |  MST  |  PSWB |  POL  |   UD  | NMSE Ave. |
> |:--------:|:-----------------:|:-----:|:-----:|:-----:|:-----:|:---------:|
> | LLaMA-7B |       0-shot      | 0.079 | 0.115 | 0.281 | 0.072 |     0.137 |
> |          |       1-shot      | 0.154 | 0.109 | 0.533 | 0.250 |     0.261 |
> |          |       2-shot      | 0.118 | 0.117 | 0.271 | 0.334 |     0.210 |
> |          |       5-shot      | 0.165 | 0.109 | 0.209 | 0.103 |     0.147 |
> |  ChatGPT |       0-shot      | 0.222 | 0.133 | 0.106 | 0.216 |     0.169 |
> |          |       1-shot      | 0.211 | 0.265 | 0.257 | 0.237 |     0.243 |
> |          |       2-shot      | 0.174 | 0.275 | 0.280 | 0.263 |     0.248 |
> |          |       5-shot      | 0.174 | 0.299 | 0.284 | 0.282 |     0.259 |
> |  ViLLaMA |       0-shot      | 0.061 | 0.059 | 0.139 | 0.140 | **0.099** |
>
> < Opinion Prediction (ESS) Few-Shot with CoT Results >
> |   Model  | Number of example |  MST  |  PSWB |  POL  |   UD  | NMSE Ave. |
> |:--------:|:-----------------:|:-----:|:-----:|:-----:|:-----:|:---------:|
> | LLaMA-7B |       0-shot      | 0.066 | 0.040 | 0.573 | 0.056 |     0.184 |
> |          |       1-shot      | 0.122 | 0.069 | 0.769 | 0.066 |     0.257 |
> |          |       2-shot      | 0.103 | 0.069 | 0.432 | 0.109 |     0.178 |
> |          |       5-shot      | 0.195 | 0.092 | 0.084 | 0.161 |     0.133 |
> |  ChatGPT |       0-shot      | 0.611 | 0.301 | 0.302 | 0.198 |     0.353 |
> |          |       1-shot      | 0.172 | 0.222 | 0.239 | 0.219 |     0.214 |
> |          |       2-shot      | 0.668 | 0.249 | 0.238 | 0.230 |     0.346 |
> |          |       5-shot      | 0.186 | 0.264 | 0.243 | 0.239 |     0.233 |
> |  ViLLaMA |       0-shot      | 0.061 | 0.059 | 0.139 | 0.140 | **0.099** |
>
> We found that ViLLaMA with VIM applied achieved a significantly lower average normalized mean squared error (NMSE) of 0.099 compared to both few-shot and few-shot CoT settings.
> In the few-shot experiments, both LLaMA-7B and ChatGPT showed their best performance in the zero-shot setting. In the few-shot CoT experiments, LLaMA-7B showed the best performance with 5-shot, while ChatGPT performed best with 1-shot.
>
> Furthermore, we conducted significance tests to verify the effectiveness of VIM. In the case of Few-shot, we compared ViLLaMA's results against LLaMA-7B's 0-shot results, and in the case of Few-shot CoT, we compared against ViLLaMA's results with LLaMA-7B's 5-shot results. We adopt a paired t-test that shows how much the method affects the results.
> For the Few-shot and Few-shot CoT, ViLLaMA's improvement over LLaMA-7B is statistically significant (p-value < 0.001).
>
> For PVQ task, we conducted experiments with the version known for having a larger number of questions and being more accurate, which consists of 40 questions. However, the dataset we have is based on a 21-question version, which has a lower number of questions and lower accuracy. In this case, since we would have had to arbitrary assign answers to the remaining questions, we were unable to conduct few-shot experiments.
> The Valuenet task is answering with "agree" or "disagree" regarding whether the model would engage in the same action as a specific value-related scenario. Our evaluation focuses not on individual answers, but on assessing the percentage of "agree" or "disagree". Similar to PVQ task, there is a challenge of assign arbitrary answers for few-shot examples, so we were unable to conduct few-shot experiments.
>
> We will use an extra page to present the results of these experiments as described above if accepted.
> - - -
> >  More ethical discussions are needed. I can see the value of the task, however, it can also create societal harm as you are predicting public opinions and values. Marginalized groups’ values may not be well representative in the data and it would be good to see the distribution regarding different groups of people.
>
> We share the sentiment that the opinions of marginalized groups should not be misrepresented or ignored. Given the lack of metadata, however, we are unable to identify members of marginalized groups in our dataset. Thus, unfortunately, we cannot empirically show that their opinions are not ignored. However, we posit that their opinions are not ignored indeed. This is because members of a marginalized group would have various value distributions, as the Schwartz value theory is designed to capture various aspects of human values. (In fact, to think that the members of a particular group share similar value distributions across all ten dimensions of Schwartz value theory would be a stereotype of the group and should be avoided.) Following this logic, members of a marginalized group would have distinct value distributions, and the prediction of each member’s opinion would be affected by the (hypothetical) group of people sharing similar value distributions. Note, the members of this hypothetical group may belong to various groups, not a particular marginalized group in real life.
>
> Still, we think you brought up an important point, and we will include more ethical discussions in the paper.

---

### Meta-Review · Area_Chair_rpuc · 2023-09-19

**Recommendation:** 4

**Metareview:**

The paper introduces a method for injecting human values into large language models (LLMs) and focuses on predicting opinions and behaviors. The reviewers find the paper well-written (uZ69, FesA) and the topic of research exciting and timely topics (uZ69). Most reviewers acknowledge the novelty and interest in the paper's idea of injecting human values into LLMs and its potential societal impact. Reviewers vyUS and FesA commend the extensive experiments conducted to demonstrate the method's efficacy. Reviewer uZ69 appreciates the well-written and clear presentation of the paper and the sound experimental setting.

The reviewers have highlighted some concerns and suggestions. Reviewer vyUS raises concerns about the performance of the VILLAMAag model compared to the baseline and suggests a more in-depth analysis of the labeling results. They also recommend ethical discussions regarding potential societal harm. Reviewer FesA notes that the paper's novelty is limited, mainly relying on fine-tuning, and suggests the need for proposing novel techniques. Reviewer 1Yni suggests refining the expression of the contributions, conducting significance tests to verify further the effectiveness of the Value Injection Method (VIM), and including hyper-parameter details for LLMs.

In summary, the reviewers collectively acknowledge the potential of the paper's approach. To further improve the article, address concerns related to performance analysis and include more experimental details to help future paper readers. Also, add a discussion about the ethical implications of the proposed approach.

---

### Decision · Program_Chairs · 2023-10-07

**Decision:**

Accept-Main

**Comment:**

The paper introduces a method for injecting human values into large language models (LLMs) and focuses on predicting opinions and behaviors. The reviewers find the paper well-written (uZ69, FesA) and the topic of research exciting and timely topics (uZ69). Most reviewers acknowledge the novelty and interest in the paper's idea of injecting human values into LLMs and its potential societal impact. Reviewers vyUS and FesA commend the extensive experiments conducted to demonstrate the method's efficacy. Reviewer uZ69 appreciates the well-written and clear presentation of the paper and the sound experimental setting.

The reviewers have highlighted some concerns and suggestions. Reviewer vyUS raises concerns about the performance of the VILLAMAag model compared to the baseline and suggests a more in-depth analysis of the labeling results. They also recommend ethical discussions regarding potential societal harm. Reviewer FesA notes that the paper's novelty is limited, mainly relying on fine-tuning, and suggests the need for proposing novel techniques. Reviewer 1Yni suggests refining the expression of the contributions, conducting significance tests to verify further the effectiveness of the Value Injection Method (VIM), and including hyper-parameter details for LLMs.

In summary, the reviewers collectively acknowledge the potential of the paper's approach. To further improve the article, address concerns related to performance analysis and include more experimental details to help future paper readers. Also, add a discussion about the ethical implications of the proposed approach.